# Comprehensive Phytochemical Profiling, Biological Activities, and Molecular Docking Studies of *Pleurospermum candollei*: An Insight into Potential for Natural Products Development

**DOI:** 10.3390/molecules27134113

**Published:** 2022-06-26

**Authors:** Maqsood Ahmed, Kashif-ur-Rehman Khan, Saeed Ahmad, Hanan Y. Aati, Chitchamai Ovatlarnporn, Muhammad Sajid-ur Rehman, Tariq Javed, Anjum Khursheed, Bilal Ahmad Ghalloo, Rizwana Dilshad, Maryam Anwar

**Affiliations:** 1Department of Pharmaceutical Chemistry, Faculty of Pharmacy, The Islamia University of Bahawalpur, Bahawalpur 63100, Pakistan; dr.maqsoodrao@gmail.com (M.A.); rsahmed_iub@yahoo.com (S.A.); anjumkhursheedrana@gmail.com (A.K.); drbilal29@hotmail.com (B.A.G.); rizwanadilshad8@gmail.com (R.D.); mariyama2014@hotmail.com (M.A.); 2Department of Pharmacognosy, College of Pharmacy, King Saud University, Riyadh 11495, Saudi Arabia; 3Department of Pharmaceutical Chemistry, Faculty of Pharmaceutical Sciences, Prince of Songkla University, Hat Yai 90110, Thailand; chitchamai.oo@psu.ac.th; 4Department of Pharmacognosy, Faculty of Pharmacy, The Islamia University of Bahawalpur, Bahawalpur 63100, Pakistan; sj_pharmacist@iub.edu.pk; 5Lahore Pharmacy College (LMDC), Lahore 53400, Pakistan; tjavedpk@gmail.com

**Keywords:** *Pleurospermum candollei*, natural compounds, antioxidant, antidiabetic, thrombolytic, antibacterial, pharmaceuticals, molecular docking

## Abstract

The purpose of this study was to find the biological propensities of the vegetable plant *Pleurospermum candollei* by investigating its phytochemical profile and biological activities. Phytochemical analysis was done by spectroscopic methods to investigate the amount of total polyphenols, and biological evaluation was done by the different antioxidant, enzyme inhibitory (tyrosinase, α-amylase, and α-glucosidase), thrombolytic, and antibacterial activities. The highest amount of total phenolic and flavonoid contents was observed in methanolic extract (240.69 ± 2.94 mg GAE/g and 167.59 ± 3.47 mg QE/g); the fractions showed comparatively less quantity (57.02 ± 1.31 to 144.02 ± 2.11 mg GAE/g, and 48.21 ± 0.75 to 96.58 ± 2.30 mg QE/g). The effect of these bioactive contents was also related to biological activities. GCMS analysis led to the identification of bioactive compounds with different biological effects from methanolic extract (antioxidant; 55.07%, antimicrobial; 56.41%), while the identified compounds from the *n*-hexane fraction with antioxidant properties constituted 67.86%, and those with antimicrobial effects constituted 82.95%; however, the synergetic effect of polyphenols may also have contributed to the highest value of biological activities of methanolic extract. Molecular docking was also performed to understand the relationship of identified secondary metabolites with enzyme-inhibitory activities. The thrombolytic activity was also significant (40.18 ± 1.80 to 57.15 ± 1.10 % clot lysis) in comparison with streptokinase (78.5 ± 1.53 to 82.34 ± 1.25% clot lysis). Methanolic extract also showed good activity against Gram-positive strains of bacteria, and the highest activity was observed against *Bacillus subtilis*. The findings of this study will improve our knowledge of phytochemistry, and biological activities of *P. candollei*, which seems to be a ray of hope to design formulations of natural products for the improvement of health and prevention of chronic diseases; however, further research may address the development of novel drugs for use in pharmaceuticals.

## 1. Introduction

Recently, the research on food plants and their bioactive ingredients is increasing due to the increased awareness by human consumers about the nutritional and functional properties of food ingredients, which have beneficial effects related to antioxidants and other biological activities and play an extensive role in maintaining and improving the human health [1,2]. The lower prevalence of metabolic disorders and other chronic diseases in the people of rural areas may also be attributed to the high consumption of plant foods, which are rich sources of bioactive compounds [3,4]. The plants with a high content of bioactive phytochemicals are loved as medicines due to their anti-inflammatory, antibacterial, anticancer sedative, antidepressant, anxiolytic, anticonvulsant, antispasmodic, and anti-HIV properties [5]. Laboratory-based work not only increases the knowledge of phytochemical composition but also finds the relationship between disease and the bioactive compounds [6], which leads to the development of formulations and functional foods with improved health benefits and safety from the plants [7]. The thrombotic events are also growing these days with a possible relation with the pandemic of COVID-19 and other infections in addition to numerous factors [8,9]. These pieces of evidence inspired us to select a plant for investigation that has shreds of evidence of use in traditional medicine but has not been evaluated scientifically for biological propensities nor for its bioactive compounds’ usefulness as a functional food to provide health-promoting effects and prevent human disease.

*Pleurospermum candollei* Benth. ex C.B.Clarke. is commonly known as Braq Shandun/Shoogroon/Shabdun in the Karakoram and Himalayan zones and belongs to the family Umbelliferae/Apiaceae. It is a 30–40 cm-long herb consumed as a vegetable by local tribes [10]. People of the Karakoram area use it for taste and nutritional benefits. This specie has also been used for different ailments and is available for commercial sale also in these regions. The whole plant is used to cure abdominal problems and stomach disorders. It also decreases cholesterol and blood pressure and provides relief from heart problems and gastric troubles [11,12]. One teaspoonful of dried plant powder is used with milk once a day for a week to treat headache and fever and can also be used by cooking with leafy vegetables for this purpose [13]. It is employed for the treatment of respiratory disorders, and the evidence also shows its good effects in pain, unconsciousness, and cerebral disorders. Many other ailments are also getting cured by the people of the Himalayan zone using this plant [14]. Stem powder of this plant has been used in joints problem and back pain in the area of Gilgit-Baltistan. Male and female infertility have also been treated using preparations from this plant [11]. It is also used to treat diarrhea in animals [15]. Ali et al., separated some compounds also to correlate the anti-inflammatory properties of *P. candollei* [14].

The Apiaceae family and *Pleurospermum* genus have been reported as good sources of natural antioxidants and used for their medicinal, pharmaceutical, nutraceutical, cosmaceutical, and food value due to the presence of many bioactive phytochemicals and their versatile biological activities [16,17]. *P. candollei*, despite the shreds of evidence of traditional uses for health benefits, has not been thoroughly investigated for its phytochemical potential and biological activities, therefore, its use as a source of antioxidants and other bioactive compounds for nutraceuticals and pharmaceutical formulations as well as functional foods, is still unknown. So the current study was designed to explore the chemical profile via total bioactive contents and GCMS analysis; a comprehensive biological screening was performed through antioxidant, enzyme-inhibitory (along with in silico studies), antibacterial, and thrombolytic activities to disclose ingredients for nutraceuticals, functional foods, and pharmaceutical possibilities. This study may also help to provide a rationale for the use of *P. candollei* in traditional medicinal system.

## 2. Results and Discussion

### 2.1. Determination of Bioactive Phytochemicals

#### 2.1.1. Quantification of Bioactive Contents (TPC and TFC)

Many researchers are interested in investigating herbs because of their phenolic phytochemicals with unique carbon scaffolds. It substantiates the importance of measuring total bioactive compounds (phenolic contents) to authenticate the use of various plants in formulating new nutraceuticals/pharmaceuticals and functional foods. It substantiates the importance of measuring total bioactive compounds (phenolic contents) to authenticate the use of various plants in formulating new nutraceuticals/pharmaceuticals and functional foods [18]. 

So the bioactive contents of *P. candollei* were determined by quantitative methods of total phenolic contents (TPC) and total flavonoid contents (TFC). Fractions of increasing polarity were used in this experiment and the results are shown in Table 1. In the present work, we used the solvents of increasing polarity (*n*-hexane, chloroform, and *n*-butanol) to make fractions of different polarities from crude extract (methanolic extract). We used the methanol (80%) to prepare the crude extract because phenolic and flavonoid compounds can efficiently be extracted using a methanol solvent, as proved by the literature [19,20]. However, separation of the extract based on polarity results in the clustering of different bioactive phytochemicals. It gives a better understanding of different behaviors of biological effects in different extracts and facilitates the bio-guided isolation of compounds also for use in pharmaceuticals and biomedicine [21]. According to the results of our study, the quantity of phenolic and flavonoid contents was highest in PCME (240.69 ± 2.94 mg GAE/g dried wt. and 167.59 ± 3.47 mg QE/g dried wt.), and their quantity decreased with the polarity of the fraction. PCHF (non-polar fraction) had the least phenolic and flavonoid contents (TPC = 57.02 ± 1.31 mg GAE/g dried wt., and TFC = 48.21 ± 0.75 mg QE/g dried wt.), while PCBF had the highest contents (TPC = 144.02 ± 2.11 mg GAE/g dry wt. and TFC = 96.58 ± 2.30 mg QE/g dried wt.). The pieces of evidence also support our results by the findings that phenolic compounds more effectively recovered with solvents of higher polarity [22,23]. Herein, it is clear from obtained data that the methanolic extract and *n*-butanol fraction had higher bioactive contents in the form of TPC and TFC, and the bioactive content’s potential of the least polar fraction was least but significant. To the best of our knowledge, no previous research has been done on the tested species of Pleurospermum to obtain data regarding its bioactive phytochemicals. Our findings were also in agreement with a recent study in which Al-Dalahmeh et al., reported that the highly polar fractions have higher levels of flavonoid and phenolic contents [24].

#### 2.1.2. Characterization of Bioactive Compounds by GCMS Analysis

GC-MS (gas chromatography-mass spectrometry) analysis is preferable for non-polar as well as for volatile compounds, and methanolic extract contained polar and non-polar compounds; moreover, the *n*-hexane fraction mainly contained volatile and non-polar compounds [25]. So methanolic extract (PCME) and *n*-hexane fraction (PCHF) of *P. candollei* were subjected to gas chromatography-mass spectrometry (GC-MS) in the search for non-polar and volatile bioactive compounds (Figure 1 and Figure 2, respectively). For the identification of natural compounds of the plant, the NIST library was used and 22 natural compounds were tentatively identified from PCME and presented in Table 2, whereas 51 natural compounds were identified from PCHF and shown in Table 3, along with their retention time in minutes, compound name, molecular formula, molecular weight, chemical class, and the percent peak area. Major constituents of PCME extract were the terpenoids, which constitute 82.61% of the non-polar components of the extract; hopanoids were the main subclass of the terpenoids; while the other compounds were steroids, alcohols, and flavonoids. The PCHF fraction mainly contained the saturated and unsaturated fatty acids (16.62%), and their esters (40.73%). Other classes of bioactive compounds were phenolic compounds (15.69%), coumarins (7.9%), steroids (4.64%), ester of fatty alcohols (2.72%), and terpenoids (1.55%), flavon (1.26%), and alkenes and alkynes (1.06%). Whereas the class of compounds present in minor quantity (less than 1% each) was alcohols (0.52%), phenolic acids (0.09%), and pyridine derivatives (0.49%). Similar results have been reported from the essential oil of a closely related species *Pleurospermum amabile*, in which Wangchuk et al. revealed many phenolics, terpenoids, and other bioactive compounds of pharmacological interest [26]. Major pharmacological activities of identified phytochemicals from *P. candollei* extract were given in Table 2. Those of the *n*-hexane fraction were presented in Table 3, along with their reference from the literature.

### 2.2. Biological Activities

#### 2.2.1. Antioxidant Activities

The antioxidant quality of a herbal drug or any plant represents its value as a food/drug supplement, which may prevent the oxidative damage of biological molecules in the human body [85]. As phenolic contents have strong antioxidant properties [86], so they are important components of natural products, nutraceuticals, pharmaceuticals, and functional foods [87], and the formulations rich in phenolic compounds reduce the risk of disease by decreasing oxidative stress [88] through a mechanism where the hydroxy group on the aromatic ring acts as an electron donor and is directly involved in the quenching of free radicals [89]. 

Data of our experiment revealed a good quantity of phenolic and flavonoid contents (Table 1), so we investigated the antioxidant potential of *P. candollei* from its methanolic extract and different fractions. Results of DPPH and ABTS assays showed (Figure 3 and Table 4) that the methanolic extract and polar fractions have more radical quenching effects compared to a non-polar fraction. The antioxidant values of the PCME were highest (165.06 ± 3.03 and 253.09 ± 2.11 mg TE/g dry weight for DPPH and ABTS assays respectively), while the antioxidant property of *n*-butanol fraction (PCBF) was found to be 101.95 ± 1.04 and 181.29 ± 1.91 mg TE/ g dry weight for DPPH and ABTS assays, respectively. The *n*-hexane fraction (PCHF) had the lowest value (DPPH = 58.09 ± 1.54, and ABTS = 113.41 ± 1.06). The higher values of antioxidant activities of methanolic extract also correlated with the antioxidant compounds (1, 2, 4, 8, 9, 10, 11, 16, 17, 19, 23, 25, 26, 27, 30, 31, 42, 43, 44, 47, 51, from Table 2), which constituted 67.86% of non-polar compounds of the total extract (Table 2); however, the highest phenolic contents also contributed to the higher antioxidant activities of PCME [90]. Moreover, the radical scavenging effect of PCHF was also significant, which may be due to the presence of many compounds (1, 2, 4, 5, 8, 9, 10, 11, 17, 19, 22, 26, 27, 29, 30, 42, 43, 44, 47, and 51, from Table 3) possessing antioxidant activities, as proven by many pieces of researches (Table 3). Those bioactive natural compounds (with antioxidant properties) contributed to 63.46% of the total PCHF fraction. So it depicted the significant antioxidant results in the PCHF fraction. Our findings were also consistent with the results of a previous study in which radical scavenging effects decreased in fractions of lesser polarity [91], this might be due to the synergistic effect of phenolic compounds in more polar fractions and they were collectively present in PCME.

The antioxidant effect was also determined by using reducing power assays like CUPRAC and FRAP. Again, the PCME showed the highest antioxidant activity (for CUPRAC; 508.32 ± 4.05, FRAP; and 361.59 ± 3.52 mg TE/g dry weight, respectively), while the activity decreased with polarity and the following pattern in the results was observed as PCBF > PCCF > PCHF. Similarly, the antioxidant activity by the FRAP method was also strongest in PCME and weakest in PCHF extract. The high reducing power may also be attributed to the presence of a higher quantity of antioxidant polyphenols and other bioactive compounds in more polar fractions [90] of *P. candollei*. However, the identified antioxidant compounds presented in Table 3 might be contributing to the significant reducing power of PCHF, while the highest antioxidant activity of PCME may be due to the synergetic effects of polyphenols in addition to the identified bioactive secondary metabolites presented in Table 2.

Total antioxidant capacity was measured by phosphomolybdenum assay and it was found that PCME has the highest antioxidant capacity (2.83 ± 0.56 mmol TE/g dry extract). However, fractions also showed significant TAC; *n*-butanol (PCBF), chloroform (PCCF), and *n*-hexane (PCHF), when compared to each other; they were: 1.85 ± 0.45, 1.80 ± 0.08, and 1.50 ± 0.25 respectively. Although, the results of PCHF showed lesser total antioxidant capacity, they were not far behind the other fractions in this regard. In addition, evidence is also present in favor of mid to non-polar volatile organic compounds, which have also shown therapeutic benefits of antioxidants relevant to traditional medicine and pharmaceuticals [92]. The findings of our study are further supported by the high number of polyphenolic compounds (Table 1) and antioxidant compounds identified by GCMS (Table 2 and Table 3).

Chelation therapy is gaining importance with the increase in pollution with heavy metals. Herbal drugs have phytochemicals to detoxify those metallic ions [93]. Although iron is essential for life it may produce free radicals, leading to oxidative damage, and its chelation prevents oxidation [94]. The metal-chelating antioxidant effect of the PCME was highest (95.63 ± 2.18 mg EDTAE/g dry weight), and the metal-chelating effect of fractions was found as PCHF = 30.22 ± 0.51, PCCF = 81.22 ± 1.01, and PCBF = 84.22 ± 1.01 mg EDTAE/g of dry weight of the fraction. The metal-chelating activity of polar fractions might be due to the presence of polyphenolic compounds [95]. The behavior of the PCHF fraction as a metal-chelating agent may be due to the presence of terpenoids [96] and other bioactive compounds (identified by GCMS and shown in Table 3), and there may be a synergetic antioxidant effect among the polyphenols (Table 1) and terpenoids, along with other bioactive natural compounds (Table 2), identified from GCMS in this study. Human health can be maintained by exploring herbs and developing formulations rich in phenolic phytochemicals, showing antioxidant properties [97,98], and *P. candollei* revealed bioactive phytochemicals and antioxidant activities in the current investigation, which provide a rational for the health benefits of this plant.

#### 2.2.2. In Vitro Enzyme Inhibition Activities

Enzyme inhibition may be an important technique in therapeutics, as various enzymes have a vital role in the physiology of human organs, and they are also involved in the etiology of different diseases. The bioactivity of tested enzymes in this study was linked with different diseases. Enzymes like α-amylase and α-glucosidase are associated with type 2 diabetes, and tyrosinase is involved in skin hyperpigmentation [99]. We presented the results of inhibition studies for these enzymes in Table 5.

Plant-derived antioxidants and tyrosinase inhibitors have gained prime importance in the cosmaceutical industry and natural products, as they have functional ingredients protecting skin from pigmentation, aging, and other skin disorders [100]. Various side effects of chemically synthetic skin remedies, like hydroquinone, made herbal skin remedies of prime importance [101]. In the current study, we already reported the good antioxidant potential of *P. candollei* that is reflectde in the tyrosinase inhibitory effect also. The highest tyrosinase inhibitory activity was revealed in methanolic extract (112.29 ± 2.79 mg KAE/g of dry weight), and tyrosinase inhibition of fractions was observed in the range of 52.61 ± 1.26 to 90.15 ± 2.10 mg KAE/g of dry weight (Table 5). A correlation between phenolic compounds, antioxidant effects, and tyrosinase inhibition was found in an Indian study [102], which also substantiates the findings of our current work. Herein, we observed that the PCME has the highest tyrosinase inhibition followed by the PCBF, PCCF, and PCHF, respectively, which may be due to the presence of different antioxidant and anti-inflammatory compounds present in high quantities (Table 2) in PCME and the synergetic effect of phenolic and flavonoid groups of compounds with varying biological potential [103]. Lup-20(29)-en-3-ol, acetate, (3β)- has been reported for tyrosinase inhibition [46], and we identified this compound in significant quantity (2.38%) as presented in Table 2, which further strengthened our findings of tyrosinase inhibition by PCME. Moreover, the *n*-hexane fraction substantially suppressed the activity of the tyrosinase enzyme due, to the identified antioxidant phytochemicals (Table 3), and evidence from the literature also favors the effect of antioxidants as melanin inhibitors [104]. A correlation was also found between DPPH antioxidation and the tyrosinase-inhibitory effect of plants. In addition to the radical scavenging mechanism, the identified compounds with anti-inflammatory (Table 2 and Table 3) effects may also be the contributors to skin-whitening effects that are much desired in Asian countries [105]. So, our data favor the *P. candollei* as a potential source to provide the bioactive and antioxidant ingredients for cosmeceuticals or the pharmaceutical industry. 

Currently, modernization, aging, and lifestyle changes are the main factors, which increase oxidative stress and lead to the development of metabolic disorders like type 2 diabetes mellitus (characterized by hyperglycemia) [106], ultimately resulting in impaired quality of life [107]. The digestive enzymes (α-amylase and α-glucosidase) are responsible for converting the ingested carbohydrates into glucose and lead to the systemic hyperglycemia. So, inhibitors of these enzymes are the antihyperglycemic agents available among the best treatment options for hyperglycemia, which control carbohydrate digestion in the intestine. However, these drugs are associated with many unwanted effects. So, there is a need for a more selective and safer agent for the induction of satisfactory therapeutic effects [108]. In an attempt to find that effect, we evaluated the fractions of *P. candollei* against α-glucosidase and α-amylase activity. Herein, the PCME showed good results for the inhibition of α-amylase and α-glucosidase (0.93 ± 0.13 mmol ACAE/g) as shown in Table 5. Among the studied fractions, PCBF had the highest activity of α-amylase inhibition (0.81 ± 0.05 mmol ACAE/g), while the other two fractions have lesser but significant activities (PCCF = 0.69 ± 0.06 and PCHF = 0.53 ± 0.08 mmol ACAE/g) compared with PCBF. For α-glucosidase-inhibitory activity, again the PCME was more effective with an inhibition value of 1.88 ± 0.15 mmol ACAE/g, respectively, while the results of other fractions showed that PCBF was more effective (0.95 ± 0.09 mmol ACAE/g), and a decrease in activity was observed with the decrease in polarity of fraction (PCCF = 0.78 ± 0.04 > 0.46 ± 0.01 mmol ACAE/g). Our finding was consistent with the reported evidence in literature in which an intimate positive correlation was found between TPC and TFC and in vitro antioxidant activity and various enzyme inhibition assays [109]. Moreover, some compounds were identified with antioxidant and anti-inflammatory effects by GCMS analysis of PCME and PCHF, which also contribute to the antidiabetic activity of the plant [110]. Evidence from the literature proved that many plant extracts and fractions show antidiabetic effects, and some plants may show antidiabetic activity even more so than the standard drug (acarbose) [111]. Although PCME showed moderate inhibition and fractions of *P. candollei* possess mild inhibition of digestive enzymes (α-glucosidase and α-amylase) as compared to equivalents of acarbose, however, our results correlate with the study on similar specie *P. benthamii* [112]. Higher antidiabetic activity of PCME was also attributed to the identified compounds A-Neooleana-3(5),12-diene, 2’-Hydroxy-3,4,4’,6’-tetra methoxy chalcone, Friedelan-3-one, and Taraxasterol, which had been reported for antidiabetic property, and collectively they constituted 31.64% of non-polar compounds of the PCME extract (Table 2, with references). These shreds of evidence widened the scope of our work on this plant species for inhibiting the activity of tested enzymes to tackle hyperglycemia and provide health benefits.

Overall, we found significant results from enzyme-inhibition assays, which establish the effectiveness and health benefits of the methanolic extract and different fractions of *P. candollei* as a potential candidate for the natural products, nutraceuticals, and pharmaceutical industry, and it may be due to the presence of a good quantity of compounds with antioxidant and anti-inflammatory property (identified molecules), and their synergic effects may also be contributing to different biological activities.

#### 2.2.3. Thrombolytic Activity

The current pandemic of COVID-19 has caused a drastic effect on the healthcare system and global economy, and the vaccination of COVID-19, although reducing the death rate, may cause thrombosis (which is also a complication of many other infections) [9,113]. Thrombosis has gained attention as a deadly complication of respiratory viral infections also (like influenza and coronavirus) [8]. Thrombosis may lead to myocardial infarction, acute ischemic stroke, venous thromboembolism, and other cardiovascular complications [114]. Experts often recommended thromboproprophylactic medications to combat thrombotic disorders of various etiologies [115], and remarkable attempts have been done in the discovery and development of safer remedies from natural constituents, so various plant sources have been investigated for antiplatelet, anticoagulant, antithrombotic, and thrombolytic effects [116]. That is why we also investigated the clot-dissolving property of our plant.

According to the results of our study, PCME was found to have maximum thrombolytic activity in a range of 55.38 ± 1.51 to 59.85 ± 1.51% of clot lysis, which was declared good when compared with streptokinase activity (78.5 ± 1.53 to 82.34 ± 1.25% clot lysis). The following order of thrombolytic activity was observed in the fractions of *P. candollei*, PCBF > PCCF > PCHF. However, results from all fractions seemed to be significant as shown in Table 6, and the results were in agreement with previous findings reported in the literature, in which the researchers found that more polar extract showed the highest thrombolytic activity in comparison with less polar fractions [117]. It was also proven by various studies that flavonoids, tannins, alkaloids, and saponins from polar organic extracts may be responsible for the thrombolytic activity of plant extracts, and developing pharmaceutical formulations or consumption of such plants as food can prevent coronary events and stroke due to thrombolytic properties [118]. So, our plant may be a potential source for developing a formulation to reduce pill burden for the patients with high thrombotic risk if further investigated by in-vivo studies.

#### 2.2.4. Antibacterial Activity

Traditional antibiotics become less effective against pathogenic bacteria due to the potential of these pathogens to develop drug resistance with extensive use of antibacterial drugs, which leads to global health threats. So, there is a need to develop new antibacterials to combat the growth of evolving bacteria [119]. Plants are the factories of nature to synthesize the secondary metabolites of varying functions, including the defense against microbial infection and parasitic infestation [120]. Plant-derived molecules may prove to be effective as antibacterial agents or may act synergistically to enhance the efficacy of older antibiotics, consequently restoring their clinical use [121], and the triterpenoids identified by GCMS may also be contributing in the antibacterial activity of methanolic extract [122]. In search of antibacterial effects, we found a significant activity of different fractions as shown in Table 7. The significant activity of PCME and PCHF can be correlated with the presence of various metabolites identified by GCMS, having antibacterial, antifungal, antiviral, and antioxidant activities (reported in the literature) as shown in Table 2 and Table 3. It was found that 56.41% of phytochemicals were identified from PCME with antimicrobial properties and in PCHF these compounds were 82.95%, as presented in Table 2 and Table 3. Our findings were close to the observation of an analysis of *P. amabile*, which describes the activity of essential oil, against *Bacillus subtilis*, and also identified some bioactive molecules by GCMS of oil. However, they also found that polar extracts were more active compared with non-polar extract [26], which correlates with our finding that PCME shows slightly greater activity, which may be due to the synergistic effect of phenolic compounds in the PCME, in addition to the identified antibacterial and antimicrobial natural compounds of the studied plant. 

Although PCHF showed a higher percentage of identified antibacterial and antimicrobial compounds (82.95%), its activity was weaker than polar fractions due to less quantity of polyphenols (Table 1), which may potentiate the antibacterial activity if present. Later, in 2014, some phenolic compounds also isolated from *P. amabile*, which showed antibacterial activity, further attributed the greater antimicrobial activity of more polar fraction due to the presence of phenolic compounds [123,124], and phenolic compounds were already detected in higher quantity from PCME, as shown in Table 1. The highest activity was observed against *Bacillus subtilis* and the lowest against *Pseudomonas aeruginosa*, and the antibacterial effect was also increased with the increase in the concentration of tested samples. The first five strains of Table 7 are Gram-positive, which showed greater inhibition by PCME, and polar fractions (PCBF and PCCF). However, the last three strains (*Escherichia coli*, *Bordetella bronchiseptica*, and *Pseudomonas aeruginosa*) are Gram-negative, which showed the least activities. Herein, the lesser zone of inhibition may be due to the high resistance of Gram-negative bacteria due to the cell membrane made of lipopolysaccharide, which is impermeable to the non-polar (lipophilic) metabolites. Peptidoglycan is the external layer in Gram-positive strains of bacteria, which is permeable to lipophilic metabolites [125]. These findings were required to compare the antibacterial activity of the plant of study with its bioactive phytochemicals and strong antioxidants potential; however, more investigation and purification of phenolic components may lead to the development of natural antibacterial agents for functional foods, nutraceuticals, and pharmaceuticals, to improve the shelf life and to extend health benefits in those systems.

### 2.3. Molecular Docking Studies

Computational chemistry provided the opportunity to the researchers to investigate the interactions between the residues of the receptor protein and ligands. The docking technique is loaded with the tools that facilitate the understanding of interaction of a presumed rigid active site with the ligand molecule and helps in drug discovery [126]. Molecular docking studies were performed for tyrosinase, α-amylase, and α-glucosidase receptors. Binding affinities and binding interactions of compounds (identified from GCMS) with major peak area were estimated for their interaction with the active sites of tested enzymes.

#### 2.3.1. Molecular Docking against Tyrosinase Enzyme

Ten identified compounds were selected from methanolic extract (Table 2) and docked against the tyrosinase enzyme. All the docked compounds showed lower binding energies (−7.2 to −8.6 kcal/mol) with tyrosinase active site as compared to kojic acid (−6.0 kcal/mol), which predict the higher binding affinity of these ligands as compared to kojic acid (Table 8). Molecular docking of the tyrosinase receptor was also performed by selecting six identified compounds from n-hexane fraction, and results were calculated as binding energies in kcal/mol (Table 8).

The following compounds from n-hexane fraction show greater affinities (lower binding energy); 2-chloroethyl linoleate (−6.6 kcal/mol); 9, 12-octadecadienoic acid (Z, Z) (−6.6 kcal/mol); and apiol (−6.1 kcal/mol) as compared to kojic acid (−6.0 kcal/mol). The higher negative value of binding energy showed the greater affinity of the ligand with tyrosinase active sites. For the further validation of the results, the ligands were docked again with the tyrosinase by using Autodock-1.5.6, and the same results were obtained in terms of the binding energy values. This study revealed that the good inhibition of tyrosinase by methanolic extract (as shown by experimental work) may be due to the higher binding affinities of docked ligands compared with kojic acid. Binding interactions of these compounds also revealed the binding forces with residues of tyrosinase active site. Different bonding interactions were involved, including hydrogen bonding, van der Waals interactions, and π bonding interactions, and some diagrammatic presentations are given in Figure 4.

#### 2.3.2. Molecular Docking against α-Amylase, and α-Glucosidase

The highest anti-diabetic activity was observed from experimental work on methanolic extract of our plant. We docked the major identified compounds from GCMS, and it revealed that all the compounds had a greater negative value of binding energy (−7.9 to −10.9 kcal/mol), which depicted higher binding affinities than acarbose (−7.7 kcal/mol) for α-amylase inhibition. Binding energies were presented in Table 8. The docking of ligands from *n*-hexane fraction resulted in lower binding affinities (binding energy; −5.0 to −6.1 kcal/mol) compared to acarbose (−7.7 kcal/mol) for α-amylase inhibition as shown in Table 8. In the case of α-glucosidase, we observed the same pattern as in the case of α-amylase that the selected ligands from methanolic extract (Table 2) showed greater affinities (binding energy; −8.4 to −9.1 kcal/mol) than standard ligand acarbose (binding energy; −7.0 kcal/mol), so the binding energies of docked ligands were lower than the standard drug, which showed higher affinity of docked ligands with active sites of α-glucosidase, and stronger antidiabetic activity was also shown by the methanolic extract in our experimental work. Herein, the compounds selected from *n*-hexane extract showed lesser binding affinities due to their higher binding energies (−5.1 to −5.9 kcal/mol) compared to the standard drug acarbose (−7.0), as shown in Table 8, and similar results were also found by experimental work. The results of molecular docking against the digestive enzymes of hyperglycemic responsibility helped us to understand the mechanism of mild to moderate results of antidiabetic activity from our experimental work for the inhibition of those enzymes. Figure 5 showed the binding interactions of α-amylase with the selected ligands, identified from GCMS, while Figure 6 showed the interaction of selected ligands with α-glucosidase enzyme.

## 3. Materials and Methods

### 3.1. Plant Collection and Authentication

The aerial parts of *P. candollei* were collected in July 2021 from Astore, Gilgit Baltistan, Pakistan. The plant was authenticated by the Herbarium of Hazara University Mansehra, Pakistan, with collection number 516 and voucher number 1885. 

### 3.2. Extraction and Fractionation

The plant was air-dried in shade and macerated in a hydroalcoholic solvent. The solvent system contained aqueous methanol (80%). This solvent system was employed as it has the property of efficient extraction of phenolic and flavonoid phytochemicals. Powdered plant material (2.5 kg) was macerated in a volume of 8 L solvent. The extract was decanted and marc squeezed. Then the obtained liquid was filtered and dried by a rotary evaporator (Heidolph, Schwabach, Germany) at 42 °C temperature and at reduced pressure. Finally, a semi-solid residue (408 g) was obtained with subsequent drying. Furthermore, fractionation of the extract was done with the help of a separating funnel and using the solvents of increasing polarity (n-hexane, chloroform, and n-butanol). Fractions were further consolidated by employing a rotary evaporator and air-dried to obtain dry material for further analysis [127].

### 3.3. Determination of Bioactive Phytochemicals

#### 3.3.1. Quantification of Bioactive Contents (TPC and TFC)

The total phenol content (TPC) of fractions was estimated quantitatively by Folin-Ciocalteau (FC reagent) method as already mentioned in the literature, with some changeovers [128]. Three readings were taken for the total phenolic contents and recorded as mg of gallic acid equivalents/gram of dried weight of fraction (mg GAE/g wt.). The total flavonoid contents (TFCs) of fractions were measured with the previously defined method with some adaptations [129]. The experiment was performed in triplicates, and total flavonoid contents were noted in mg of quercetin equivalents/g of the dried weight of fraction (mg QE/g wt.).

#### 3.3.2. Characterization of Bioactive Compounds by GCMS Analysis

The PCME extract and PCHF fraction of *P. candollei* were subjected to GC-MS analysis by following a procedure mentioned in the literature. The Percent peak area for each compound was calculated from the whole area of peaks [130].

### 3.4. Biological Activities

#### 3.4.1. Antioxidant Activities

The antioxidant potential of *P. candollei* was tested by six different methods, which include DPPH, ABTS, FRAP, CUPRAC, and TAC (phosphomolybdenum assay) by previously described methods [131,132]. The results were expressed as equivalent to Trolox, whereas the metal-chelating property was expressed as equivalent to ethylene diamine tetraacetic acid (EDTA), and all the readings were taken in triplicates. Moreover, the IC_50_ and EC_50_ values were also calculated due to the good antioxidant results of the extract and fractions (equivalent to the standard antioxidants).

#### 3.4.2. In Vitro Enzyme Inhibition Activities

The enzyme-inhibitory potential of all fractions was tested against tyrosinase, α-amylase, and α-glucosidase by using the methods reported in the literature [132]. Tyrosinase inhibition was measured as kojic acid equivalent, and inhibition of enzymes, responsible for hyperglycemia (α-amylase and α-glucosidase) was observed by acarbose equivalent potential of the extract, and its fractions and three readings were taken for each assay.

#### 3.4.3. Thrombolytic Activity

For thrombolytic activity, the blood samples were obtained from healthy volunteers by following the guidelines of the ethical committee of the Islamia University of Bahawalpur. Five volunteers were included in the study who had not been using antidepressants, oral contraceptives, or anticoagulants. Analysis was performed by taking 5.0 mL of water (sterile) and dropped into the commercially available lyophilized streptokinase (SK) injection (15,000,000 i.u.). Then it was used as a positive control for thrombolytic activity. After that, 100 μL of solution of the SK (30,000 I.U) was for thrombolysis. In the Eppendorf tubes (after weighing the tubes), 500 μL of the venous blood (from a volunteer) was taken to form a clot. The liquid (serum) was completely removed by aspiration without the disruption of the clot. The weight of the clot was measured by using the formula, weight of thrombus = weight of tube with thrombus, which is the weight of the empty tube. Finally, a 100 μL volume of sample solution (1 mg/mL) was added to each tube, and 100 μL of streptokinase in the control tube. Negative control was distilled water, which was added to each tube in a volume of 100 μL. The test tubes were incubated at 37 °C for a period of 90 minutes and weighed again after removing the fluid (to observe clot lysis). The difference in weight was calculated before and after clot lysis by the given formula and expressed as a percent of thrombolytic activity of fractions, and the experiment was performed in triplicates [133].

#### 3.4.4. Antibacterial Activity

The antibacterial effect of the plant was determined by using five Gram-positive (*Bacillus subtilis* ATCC1692, *Micrococcus luteus* ATCC 4925, *Staphylococcus epidermidis* ATCC 8724, *Bacillus pumilus* ATCC 13835, and *Staphylococcus aureus* ATCC 6538) and three Gram-negative strains (*Escherichia coli* ATCC 25922, *Bordetella bronchiseptica* ATCC 7319, and *Pseudomonas aeruginosa* ATCC 9027). These bacterial strains were procured from drug testing laboratory (DTL) Bahawalpur. Disc diffusion technique was employed to check the inhibition zones in mm. The procedure of the activity was described by previous research [134]. In disc diffusion assay, the sample of the extract/fraction or the standard (antibiotic) was diffused from a disk (loaded with sample or antibiotic) over the medium (nutrient agar), thereby creating a concentration gradient. Discs were made up of filter paper and were 5 mm in diameter. A known concentration of the sample (10, 20, and 40 mg per mL in DMSO) was impregnated on each disc, and the discs were placed on nutrient agar media (inoculated with test strains of bacteria). Co-amoxiclav in the concentration of 1 mg/mL was used as a standard antibacterial drug, and a disc without sample/standard was placed as a negative control (loaded with DMSO). The petri dishes were incubated at 37 °C for 24 h to provide optimum conditions for the growth of bacteria. A clear/distinct zone showing no bacterial growth is called “Zone of Inhibition”, and the diameter of this zone was measured in mm to determine the antibacterial activity of the sample or standard.

### 3.5. Molecular Docking Studies

In computer-aided drug designing, molecular docking is a useful technique to understand the mechanism of interaction of protein receptors with ligands. Structures of the standard ligands (kojic acid for tyrosinase and acarbose for amylase and glucosidase) were downloaded from the PubChem database in the form of SDF (structured data format) files. Biomacromolecules (tyrosinase; 10.2210/pdb3NM8/pdb, amylase; 10.2210/pdb3VX1/pdb, and glucosidase; 10.2210/pdb5ZCB/pdb) were obtained in protein data bank (PDB) format. The preparation of ligands as PDB files was done for molecular retrieval and different software tools were used, such as auto Dock vina software, Discovery Studio, PyRx, and Babel. Discovery Studio (Discovery Studio 2021 client) was used to prepare the enzyme molecules that were obtained from Protein Data Bank. Ligand molecules were identified from GC-MS and selected based on major peak areas. Babel was used for preparing ligand molecules. Vina embedded in PyRx was run by uploading prepared receptors and ligands. These structures of ligands were placed in the area of the active site of the enzyme with AutoDock vina. Outcomes of the interaction of enzyme-active site residues and ligand molecules were evaluated using the Discovery Studio Visualizer, and 2D interactions were presented in Figure 4, Figure 5 and Figure 6 [133,135].

## 4. Conclusions

The purpose of current work was to quantify and characterize the phytochemical composition and biological activities from the methanolic extract and different extraction fractions of *P. candollei*. Polyphenols and other bioactive phytochemicals were revealed by TPC, TFC, and GCMS analysis, and the identified molecules had antioxidant potential, antibacterial/antimicrobial potential, and other biological properties; moreover, biological activities of the plant were also found to be in relation to the quantity and quality of bioactive natural compounds. High values of antioxidant activities encourage the use of this plant to develop natural products for various health benefits and make it a potential source for use in natural products, pharmaceuticals, and nutraceuticals. Significant results of inhibition against tyrosinase, α-amylase, and α-glucosidase; good thrombolytic property; and antibacterial activity may also help to establish the nutritional and phyto-therapeutical role of the plant. A molecular docking study was used to explain the relationship of enzyme inhibition with the identified bioactive natural compounds. The findings of this study will not only improve our understanding of phytochemistry and biological effects of *P. candollei*, but also provide basic scientific knowledge to understand the health-improving and bioactive properties for developing the nutraceutical possibilities and natural products. However, future research may address the development of novel drugs for use in pharmaceuticals.

## Figures and Tables

**Figure 1 molecules-27-04113-f001:**
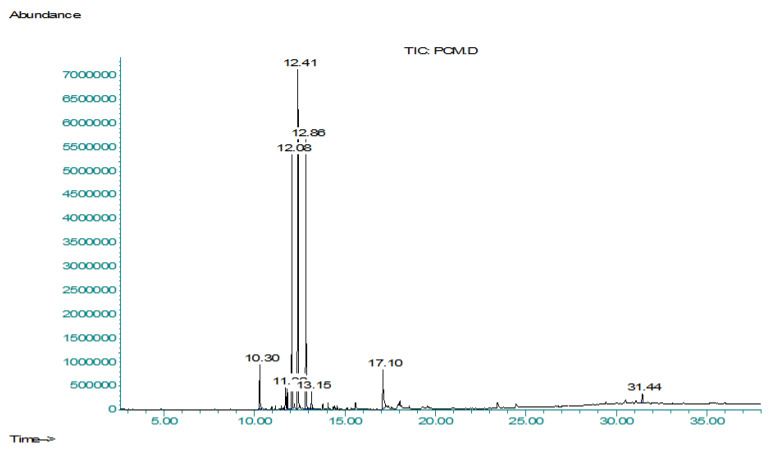
Chromatogram obtained from GCMS analysis of PCME.

**Figure 2 molecules-27-04113-f002:**
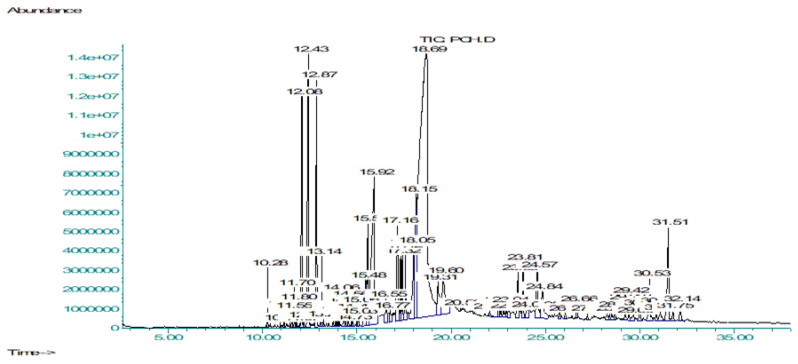
Chromatogram obtained from GCMS analysis of PCHF.

**Figure 3 molecules-27-04113-f003:**
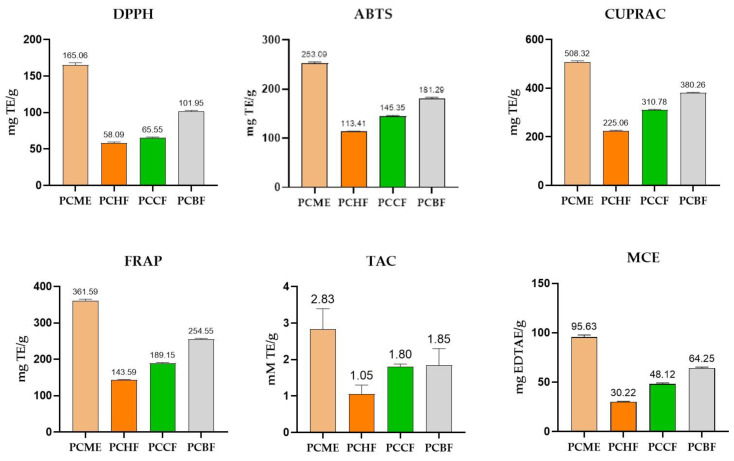
Antioxidant activities (mg and mM equivalents of standard antioxidants/g extract or fraction) of *P. candollei*.

**Figure 4 molecules-27-04113-f004:**
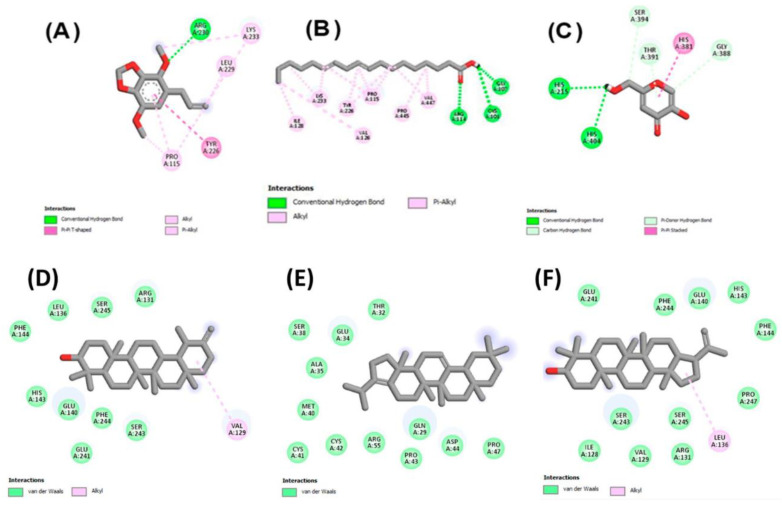
Interactions of tyrosinase active site residues with (**A**) apiol, (**B**) octadecadienoic acid, (**C**) kojic acid, (**D**) taraxasterol, (**E**) a-neooleana-3(5), 12-dien, (**F**) hopa-22(29)-ene-3alpha-ol.

**Figure 5 molecules-27-04113-f005:**
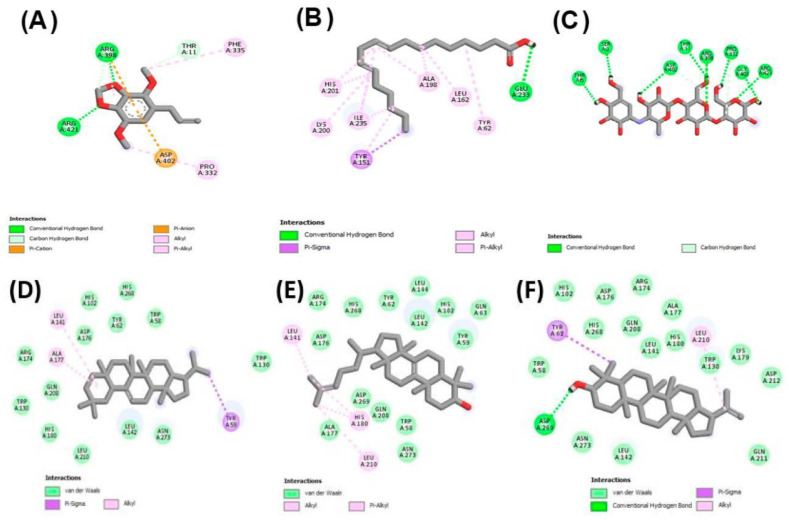
Interactions of residues of α-amylase active site (2D presentation) with (**A**) apiol, (**B**) octadecadienoic acid, (**C**) acarbose, (**D**) a-neooleana-3(5), 12-dien, (**E**) lanosterol, (**F**) 3-epimoretenol.

**Figure 6 molecules-27-04113-f006:**
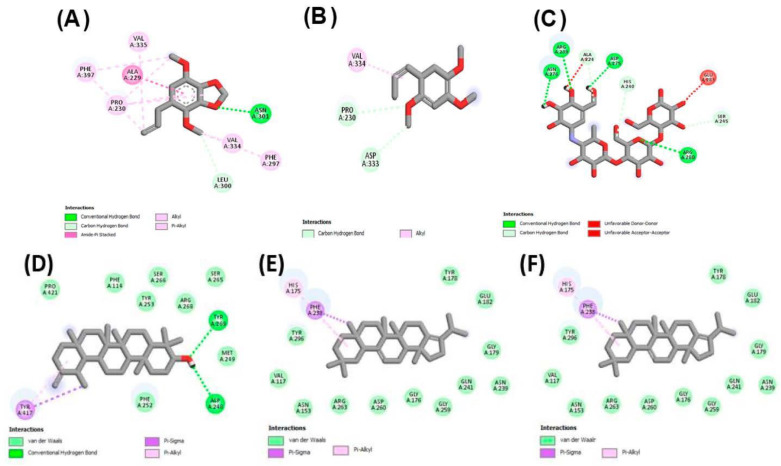
Interactions of residues of α-glucosidase active site (2D presentation) with (**A**) apiol, (**B**) asarone, (**C**) acarbose, (**D**) taraxasterol, (**E**) hopa-22(29)-ene-3alpha-ol, (**F**) a-neooleana-3(5), 12-dien.

**Table 1 molecules-27-04113-t001:** Total phenolic and total flavonoid contents (TPC and TFC) of *P. candollei* (sample conc. 1 mg/mL).

Sample Codes	TPC (mg GAE/g Dried wt.)	TFC (mg QE/g Dried wt.)
PCME	240.69 ± 2.94	167.59 ± 3.47
PCHF	57.02 ± 1.31	48.21 ± 0.75
PCCF	97.02 ± 1.83	88.32 ± 1.45
PCBF	144.02 ± 2.11	96.58 ± 2.30

Values were taken in triplicates and reported as mean ± SD. PCME: *P. candollei* methanolic extract; PCHF: *P. candollei n*- hexane fraction; PCCF: *P. candollei* chloroform fraction; PCBF: *P. candollei n*-butanol fraction; TPC: total phenolic content; TFC: total flavonoid content; GAE: gallic acid equivalent; QE: Quercetin equivalent.

**Table 2 molecules-27-04113-t002:** Phytochemical profiling of methanolic extract (PCME) of *P. candollei* through GCMS analysis.

Sr.no.	RT	Tentative Identification of Compounds	Molecular Formula	Molecular Weight	Chemical Class	Area %	Reported Activities from Literature
1	15.13	Hexadecanoic acid, methyl ester	C_17_H_34_O_2_	270.5	Fatty acid ester	0.05	Antioxidant, antimicrobial [27]
2	17.34	β-Amyrin	C_30_H_50_O	426.70	Terpenoid	2.77	Antioxidant, anti-inflammatory, antibacterial, antiulcer, antiarthritic, antidiahreal [28]
3	17.42	2(1H)Naphthalenone, 3,5,6,7,8,8a-hexahydro-4,8a-dimethyl-6-(1-methylethenyl)-	C_15_H_22_O	218.33	Terpenoid	0.45	Anticancer, antioxidant ani-inflammatory, analgesic, sedative [29]
4	18.46	4,6,6-Trimethyl-2-(3-methylbuta-1,3-dienyl)-3-oxatricyclo[5.1.0.0(2,4)]octane	C_15_H_22_O	218.33	Terpenoid	9.20	Antioxidant [27]
5	19.21	A-Neooleana-3(5),12-diene	C_30_H_48_	408.7	Terpenoid	9.92	Anti-inflammatory, antimicrobial [30], antidiabetic [31]
6	19.31	3-Epimoretenol	C_30_H_50_O	426.7	Terpenoid	3.37	Anti-inflammatory, analgesic [32]
7	19.77	9,19-Cyclolanost-24-en-3-ol, acetate, (3beta)-	C_32_H_52_O_2_	468.8	Steroid ester	8.00	Antibacterial, antioxidant [33]
8	20.27	Lanosterol	C_30_H_50_O	426.7	Steroid	1.97	Antioxidant [34], cytoprotective, neuroprotective, anti-inflammatory [35]
9	23.27	Phenanthrene, 7-ethenyl-1,2,3,4,4a,4b,5,6,7,8,8a,9-dodecahydro-1,1,4b,7-tetramethyl-	C_20_H_32_	272.5	Terpenoid	0.58	Antioxidant, antibacterial [36]
10	25.21	D:C-Friedours-7-en-3-one	C_30_H_48_O	424.7	Terpenoid	1.93	Antioxidant, anti-inflammatory, antibacterial [37]
11	25.33	9,19-Cycloergost-24(28)-en-3-ol, 4,14-dimethyl-, acetate, (3β,4α,5α)-	C_32_H_52_O_2_	468.75	Steroid ester	0.66	Anti-inflammatory, antibacterial [32]
12	25.49	2′-Hydroxy-3,4,4′,6′-tetramethoxychalcone	C_19_H_20_O_6_	344.4	Flavonoid	1.93	Antioxidant, antibacterial, antidiabetic [38]
13	25.63	A′-Neogammacer-22(29)-en-3-one	C_30_H_48_O	424.7	Terpenoid	0.47	Antibacterial, antioxidant [39]
14	26.22	Cedran-diol, 8S,14-	C_15_H_26_O_2_	238.37	Terpenoid	9.22	Anti-inflammatory, anticancer [40]-
15	26.64	Taraxasterol	C_30_H_50_O	426.7	Terpenoid	19.00	Antidiabetic [31], anti-inflammatory, analgesic [41]
16	26.93	Hop-22(29)-en-3.beta.-ol	C_30_H_50_O	426.7	Terpenoid	12.13	Antibacterial, antioxidant [42]
17	27.18	Lupeol	C_30_H_50_O	426.7	Terpenoid	10.40	Antimicrobial, antioxidant, anticancer, anti-inflammatory [43]
18	27.55	Urs-12-en-3-ol, acetate, (3beta)-	C_32_H_52_O_2_	468.8	Ester	3.31	Antioxidant, antimicrobial, anticancer [44]
19	28.21	Friedelan-3-one	C_30_H_50_O	426.7	Terpenoid	0.79	Antioxidant, antidiabetic, antimicrobial [45]
20	29.40	2,4,6-Cycloheptatrien-1-one, 3-hydroxy-	C_7_H_6_O_2_	300.31	Alcohol	0.38	-
21	30.69	Lup-20(29)-en-3-ol, acetate, (3β)-	C_32_H_52_O_2_	468.8	Terpenoid	2.38	Anti-inflammatory, analgesic, antibacterial [32], tyrosinase-inhibitory effect [46]
22	31.12	Olean-18-en-28-oic acid, 3-oxo-, methyl ester	C_31_H_48_O_3_	468.7	Fatty acid ester	1.09	Antimicrobial, antioxidant [47]

RT; Retention time in minutes.

**Table 3 molecules-27-04113-t003:** Phytochemical profiling of *n*-hexane (PCHF) fraction of *P. candollei* through GCMS analysis.

Sr.no.	RT	Tentative Identification of Compounds	Molecular Formula	Molecular Weight	Chemical Class	Area %	Reported Activities from Literature
1	10.28	Methyl iso-eugenol 2	C_11_H_14_O_2_	178.22	Phenolic	0.55	Antibacterial, antioxidant. [48]
2	10.94	Benzene, 1,2,3-trimethoxy-5-(2-propenyl)-	C_12_H_16_O_3_	208.25	Phenolic	0.06	Anti-inflammatory, antioxidant [49] antifungal, antimicrobial [50]
3	11.15	Bicyclo[6.3.0]undec-1(8)-en-3-ol, 2,2,5,5-tetramethyl-	C_15_H_26_O	222.19	Terpenoid	0.17	Cytotoxic, antiplasmodial, antiviral, anti-inflammatory [51]
4	11.55	3-Hydroxy-4-methoxycinnamic acid	C_10_H_10_O_4_	194.18	Phenolic	0.28	Antioxidant, anti-inflammatory [52]
5	11.80	9-Anthracenecarboxylic acid	C_15_H_10_O_2_	222.24	Aromatic carboxylic acid	0.24	Antimicrobial, antifungal [53], antibacterial, anticancer [54]
6	11.97	Diepi-.alpha.-cedrene epoxide	C_15_H_24_O	220.35	Terpenoid	0.06	Cytotoxic, antibacterial [54]
7	12.08	Isoelemicin	C_12_H_16_O_3_	208.25	Phenolic	3.35	Antimicrobial [55]
8	12.20	1,3-Benzodioxole, 4,5-dimethoxy-7-(2-propenyl)-	C_12_H_14_O_4_	222.24	Phenolic	0.14	Antimicrobial, antioxidant, anticancer [56]
9	12.43	Asarone	C_12_H_16_O_3_	208.25	Phenolic	6.30	Hypoglycemic, antimicrobial, anti-Alzheimer’s disease, anticonvulsive, antiepileptic and antioxidant properties [57,58]
10	12.87	Apiol	C_12_H_14_O_4_	222.23	Phenolic	3.97	Antioxidant, antimicrobial [59]
11	13.07	2H-1-Benzopyran-2-one, 7-methoxy-	C_10_H_8_O_3_	176.17	Coumarin	0.12	Antioxidant, analgesic, anticoagulant, anti-inflammatory, antimicrobial [60].
12	13.14	Aspidinol	C_13_H_18_O_4_	238.28	Phenolic	0.59	Antibacterial [61]
13	13.22	Tetradecanoic acid	C_14_H_28_O_2_	228.37	Fatty acid	0.14	-
14	13.88	Phenol,2-[[(4-methylphenyl)imino]methyl]-	C_14_H_13_NO	211.26	Phenolic	0.15	_
15	13.97	2,4,6-Trimethoxyacetophenone	C_11_H_14_O_4_	210.23	Phenolic	0.14	Antibacterial and synergistic effect with antibiotics [62]
16	14.06	1-Methoxy-3-(2-hydroxyethyl)nonane	C_12_H_26_O_2_	202.33	Alcohol	0.31	Antifungal, antioxidant [63]
17	14.34	9-Octadecyne	C_18_H_34_	250.5	Alkyne	0.34	Larvicidal, antioxidant [64]
18	14.42	Tricyclo[7.2.0.0(2,6)]undecan-5-ol	C_15_H_26_O	222.37	Terpenoid	0.18	-
19	14.79	7,10,13-Hexadecatrienoic acid, methyl ester	C_17_H_28_O_2_	264.40	Fatty acid ester	0.17	Antioxidant, anti-inflammatory, antimicrobial [65]
20	15.03	3,4-dihydrocoumarin	C_9_H_8_O_2_	148.16	Coumarin	0.12	Anticoagulant, antifungal, anticancer, antibacterial [66]
21	15.08	Hexadecanoic acid, methyl ester	C_17_H_34_O_2_	270.45	Fatty acid ester	0.20	Antibacterial, antifungal [67]
22	15.48	11,14,17-Eicosatrienoic acid, methyl ester	C_20_H_34_O_2_	306.5	Fatty acid	1.52	Anti-microbial, anti-cancer, anti-hair fall, CNS depressant, analgesic, anti-inflammatory, antipyretic, anti-arthritic, anti-coronary [68]
23	15.92	n-Hexadecanoic acid	C_16_H_32_O_2_	256.42	Fatty acid	9.40	Antioxidant, antibacterial, anti-inflammatory [69]
24	17.16	7H-Furo[3,2-g][1]benzopyran-7-one	C_11_H_6_O_5_	218.16	Coumarin	2.42	-
25	17.32	9,12-Octadecadienoic acid, methyl ester	C_19_H_34_O_2_	294.5	Fatty acid ester	0.84	Antioxidant, anti-inflammatory, antimicrobial [65]
26	17.42	9,12,15-Octadecatrienoic acid, methyl ester	C_19_H_32_O_2_	278.4	Fatty acid ester	0.92	Antimicrobial [70] Antioxidant, anti-inflammatory, antimicrobial [65]
27	17.58	Phytol	C_20_H_40_O	296.5	Terpenoid	1.14	Antioxidant, anticancer [71]
28	18.05	Z,Z-10,12-Hexadecadien-1-ol acetate	C_18_H_32_O_2_	280.4	Ester of fatty alcohol	2.55	Antimicrobial [72]
29	18.15	9,12-Octadecadienoic acid (Z,Z)	C_18_H_32_O_2_	280.4	Fatty acid	3.55	Antibacterial, antifungal, anti-inflammatory, antineoplastic [73]
30	18.69	2-Chloroethyl linoleate	C_20_H_35_ClO_2_	342.9	Fatty acid ester	39.69	Cytotoxic, antioxidant, antimicrobial [74]
31	19.31	Flavone	C_15_H_10_O_2_	222.24	Flavonoid	1.26	Antibacterial, antiviral, antifungal, antioxidant, anti-inflammatory [75]
32	19.59	Pimpinellin	C_13_H_10_O_5_	246.21	Furocoumarin	2.57	Strong antibacterial [76]
34	22.05	Phenol, 2,2’-methylenebis[6-(1,1-dimethylethyl)-4-methyl-	C_23_H_32_O_2_	340.49	Phenolic	0.16	α-amylase inhibitor [77]
35	22.52	1,3,14,16-Nonadecatetraene	C_19_H_32_	260.45	Alkene	0.23	-
36	22.77	(R)-(-)-14-Methyl-8-hexadecyn-1-ol	C_17_H_32_O	252.4	Alcohol	0.21	-
37	22.92	1,5,9,13-Tetradecatetraene	C_14_H_22_	190.32	Alkene	0.14	-
38	23.04	9-Tricosene, (Z)-	C_23_H_46_	322.6	Alkene	0.35	-
39	23.55	4-(3-Methyl-2-oxobutoxy)-7H-furo[3,2-g][1]benzopyran-7-one	C_16_H_14_O_5_	286.28	Coumarin	1.20	Antibacterial [76]
40	24.57	7H-Furo(3,2-g)(1)benzopyran-7-one, 4,9-dihydroxy-	C_11_H_6_O_5_	218.16	Coumarin	1.47	Antibacterial, antiacetyl, and butyrylcholinesterase [76]
41	24.84	6-Acetylchrysene	C_19_H_14_	242.3	Phenanthrene	1.27	-
42	25.76	13-Tetradecen-1-ol acetate	C_16_H_30_O_2_	254.41	Ester fatty alcohol	0.17	Antibacterial, antioxidant [78]
43	27.22	3,4-Dimethoxycinnamic acid	C_11_H_12_O_4_	208.21	*Cinnamic acid*	0.09	Neuroprotactive, antioxidant, anticancer [79]
44	28.40	N-hydroxy-N’-[2-(trifluoromethyl)phenyl]pyridine-3-carboximidamide	C_13_H_10_F_3_N_3_O	281.23	Pyridine derivative	0.49	Antioxidant, anti-inflammatory, antimicrobial [80]
45	28.69	Stigmastan-6,22-dien, 3,5-dedihydro-	C_29_H_46_	394.7	Steroid	0.18	Antifungal [81]
46	29.66	Stigmastane-3,6-dione	C_29_H_48_O_2_	428.7	Steroid	0.23	-
47	30.53	Stigmasta-5,22-dien-3-ol, acetate	C_31_H_50_O_2_	454.7	Steroid	1.16	Antimicrobial, antioxidant [82]
48	30.88	Ergosta-4,6,22-trien-3.beta.-ol	C_28_H_44_O	396.6	Steroid	0.29	-
49	31.51	Clionasterol acetate	C_31_H_52_O_2_	456.7	Steroid	2.51	-
50	31.75	3β-acetoxy-pregna-5,16-dien-20-one	C_23_H_32_O_3_	298.5	Steroid	0.27	Anti-inflammatory, antibacterial [83]
51	32.14	Vitamin E	C_29_H_50_O_2_	430.7	Chromanol derivative	0.46	Antioxidant, anticancer, anti-inflammatory [84].

RT; retention time in minutes.

**Table 4 molecules-27-04113-t004:** IC_50_ values of antioxidant activities of methanolic extract and different fractions of *P. candollei* (sample conc. 1 mg/mL).

Sample Codes	Radical Scavenging Assay		Reducing Power Assay		Reducing/Metal-Chelating Assay	
	DPPH (IC_50_ mg/mL)	ABTS (IC_50_ mg/mL)	CUPRAC (EC_50_ mg/mL)	FRAP (EC_50_ mg/mL)	TAC (EC_50_ mM/mL)	MCE (IC_50_ mg/mL)
PCME	3.64 ± 0.73 ^a^	2.49 ± 0.84 ^a^	1.20 ± 0.45 ^a^	1.67 ± 0.68 ^a^	0.45 ± 0.06 ^a^	2.16 ± 0.48 ^a^
PCHF	10.34 ± 1.41 ^d^	5.31 ± 1.04 ^d^	2.67 ± 0.80 ^d^	4.20 ± 1.06 ^d^	0.24 ± 0.02 ^d^	7.00 ± 0.86 ^d^
PCCF	9.23 ± 1.95 ^c^	4.14 ± 1.21 ^c^	1.93 ± 0.79 ^c^	3.17 ± 1.10 ^c^	0.29 ± 0.08 ^c^	4.38 ± 0.62 ^c^
PCBF	5.90 ± 1.06 ^b^	3.32 ± 0.95 ^b^	1.59 ± 0.16 ^b^	2.36 ± 0.75 ^b^	0.30 ± 0.04 ^b^	3.28 ± 0.35 ^b^

Values were taken in triplicates (*n* = 3) and reported as mean ± SD PCME: *P. candollei* methanolic extract, PCHF: *P. candollei n*-hexane fraction, PCCF: *P. candollei* chloroform fraction, PCBF: *P. candollei n*-butanol fraction, DPPH: 2,2-diphenyl-1-picrylhydrazyl assay, ABTS: 2,2′-azino-bis(3-ethylbenzothiazoline-6-sulfonic acid) assay, CUPRAC: cupric reducing antioxidant capacity, FRAP: Ferric reducing antioxidant power, TAC: Phosphomolybdenum assay, MCE: metal-chelating effect. ^a, b, c, d^ Letters in one column indicate significant differences in the activities of tested extract and fractions (*p* < 0.05).

**Table 5 molecules-27-04113-t005:** Enzyme-inhibitory activities of methanolic extract and different fractions of *P. candollei* (sample conc. 1 mg/mL).

Sample Codes	Tyrosinase (mg KAE/g Dried wt.)	α-Amylase (mmol ACAE/g Dried wt.)	α-Glucosidase (mmol ACAE/g Dried wt.)
PCME	112.29 ± 2.79	0.93 ± 0.07	1.88 ± 0.15
PCHF	52.61 ± 1.26	0.53 ± 0.08	0.46 ± 0.01
PCCF	82.91 ± 1.79	0.69 ± 0.06	0.78 ± 0.04
PCBF	90.15 ± 2.10	0.81 ± 0.05	0.95 ± 0.09

Values were taken in triplicates and reported as mean ± SD. PCME, *P. candollei* methanolic extract; PCHF, *P. candollei* -hexane fraction; PCCF, *P. candollei* chloroform fraction; PCBF, *P. candollei n*-butanol fraction; KAE, kojic acid equivalent; ACAE, acarbose equivalent.

**Table 6 molecules-27-04113-t006:** Thrombolytic activity (% clot lysis) of fractions of *P. candollei* from different blood samples (subject A–E).

Sample Codes	Subject A	Subject B	Subject C	Subject D	Subject E
PCME	55.38 ± 1.51	58.16 ± 1.9	55.45 ± 1.18	58.65 ± 1.25	59.85 ± 1.51
PCHF	40.18 ± 1.80	43.1 ± 1.69	42.51 ± 0.98	43.8 ± 0.82	42.63 ± 1.35
PCCF	41.54 ± 0.95	48.15 ± 1.41	47.15 ± 1.11	50.14 ± 1.61	47.85 ± 1.80
PCBF	52.15 ± 0.68	57.25 ± 0.94	55.10 ± 1.12	56.95 ± 1.70	57.15 ± 1.10
Streptokinase	78.5 ± 1.53	80.14 ± 0.91	81.43 ± 1.39	82.34 ± 1.25	79.12 ± 2.3

Values were taken in triplicates and reported as mean ± SD. PCME; *P. candollei* methanolic extract, PCHF; *P. candollei* n-hexane fraction, PCCF; *P. candollei* chloroform fraction, PCBF; *P. candollei* n-butanol fraction.

**Table 7 molecules-27-04113-t007:** Antibacterial activity of methanolic extract and different fractions of *P. candollei*.

Strain Name	Zone of Inhibition (mm) of Standard (Co-Amoxiclav) (Concentration = 1 mg/mL)	Concentration (mg/mL)	Zone of Inhibition of PCME Extract (mm)	Zone of Inhibition of PCHF Fraction (mm)	Zone of Inhibition of PCCF Extract (mm)	Zone of Inhibition of PCBF Extract (mm)
*Bacillus subtilis*	23	10	7	-	-	7
		20	13	10	12	12
		40	18	16	16.5	18
*Micrococcus luteus*	20	10	7	6	6	6
		20	13	10	11	13
		40	17	16	17	17
*Staphylococcus epidermidis*	24	10	5	-	-	-
		20	10	8	8	10
		40	12	10	13	15
*Bacillus pumilus*	22	10	6	-	-	-
		20	11	7	9	9
		40	15	12	14	14.5
*Staphylococcus aureus*	23	10	7	-	6	6
		20	13	10	11	12
		40	15.5	12	12.5	14
*Escherichia coli*	25	10	-	-	-	-
		20	8	-	7	7.5
		40	10	6	8	9
*Bordetella bronchiseptica*	26	10	-	-	-	-
		20	8	-	6	8
		40	9	10	11	11.5
*Pseudomonas aeruginosa*	18	10	-	-	-	-
		20	-	-	-	-
		40	7	-	-	6

PCME, *P. candollei* methanolic extract; PCHF, *P. candollei n*-hexane fraction; PCCF, *P. candollei* chloroform fraction; PCBF, *P. candollei n*-butanol fraction.

**Table 8 molecules-27-04113-t008:** Binding energies of docked compounds against tyrosinase, α-amylase, and α-glucosidase.

Name of Compound	Binding Energy of Ligand with Tyrosinase (kcal/mol)	Binding Energy of Ligand with α-Amylase (kcal/mol)	Binding Energy of Ligand with α-Glucosidase (kcal/mol)
Taraxasterol	−8.6	−9.5	−8.5
beta-Amyrin	−8.3	−9.1	−8.8
Hopa-22(29)-ene-3alpha-ol	−8.0	−8.9	−9.0
A-Neooleana-3(5),12-dien	−7.9	−10.9	−8.7
Urs-12-en-3-ol, acetate, (3beta)-	−7.7	−8.8	−8.4
Lupeol	−7.4	−8.8	−9.1
Lanosterol	−7.3	−9.7	−8.7
Lup-20(29)-en-3-ol, (3beta)-	−7.3	−8.7	−9.0
3-Epimoretenol	−7.3	−8.9	−8.6
9,19-Cyclolanost-24-en-3-ol, acetate, (3beta)-	−7.2	−7.9	−8.5
Standad	−6.0 ^1^	−7.7 ^2^	−7.0 ^2^

^1^ Kojic acid and ^2^ acarbose.

## Data Availability

Not applicable.

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
