# Peer review of "Comprehensive Phytochemical Profiling, Biological Activities, and Molecular Docking Studies of Pleurospermum candollei: An Insight into Potential for Natural Products Development"

_molecules, 2022, doi:10.3390/molecules27134113_

Round 1

Reviewer 1 Report

Dear editor; The att ached articled was checked. The manuscript contain interesting information about Comprehensive Phytochemical Profiling, Biological activities, and Molecular Docking Studies of Pleurospermum candollei: An Insight into Potential for Natural Products Development

It is generally a good work. The scientific and presentation level of the manuscript is high. The manuscript is recommended for publication with minor revisions.

 In text: You can write the name of the species in short. Write the name of the authority to the end of the plant’s name

http://www.theplantlist.org/tpl1.1/record/kew-2405060

In introduction: Pleurospermum candollei Benth. ex C.B.Clarke

How does the use of the plant species compare with those reported from other neighbouring regions, or regions on the rest of the country or neighbouring countries?

I believe that a comparative comment on the relevance of the different plants in world or elsewhere's places would be good.

Botanical studies have considerably increased in recent years :

A.M.A. Kawarty, A. M., Behçet, L. & Cakilcioğlu, U. An ethnobotanical survey of medicinal plants in Ballakayati (Erbil, North Iraq). Turk. J. Botany 44, 345–357 (2020).

Satıl, F. & Selvi, S. Ethnobotanical Features of Ziziphora L. (Lamiaceae) Taxa in Turkey. International Journal Nature and Life Sciences. 4, 56-65. (2020).

Herbs are used for medical purposes. That herbs are used for medical purposes may be added in the introduction part of the article. 

The paper should be edited according to the writing rules of the journal.

The references do not conform with the journal style.

Author Response

Response to Respected Reviewer 1 comments

Dear editor; The attached article was checked. The manuscript contain interesting information about Comprehensive Phytochemical Profiling, Biological activities, and Molecular Docking Studies of Pleurospermum candollei: An Insight into Potential for Natural Products Development

It is generally a good work. The scientific and presentation level of the manuscript is high. The manuscript is recommended for publication with minor revisions.

Thank you for your kind and very keen observation. Also, thank you so much for your valuable insight on improving the manuscript. We greatly appreciate the reviewers’ comments and constructive suggestions. Those valuable comments are constructive to improve our manuscript and provide meaningful guidance for our future research. We hope this revised manuscript will meet the satisfaction of the reviewer.

In text: You can write the name of the species in short. Write the name of the authority to the end of the plant’s name. In introduction: Pleurospermum candollei Benth. ex C.B.Clarke.

Response: Included full name of plant according to your suggestion.

How does the use of the plant species compare with those reported from other neighbouring regions, or regions on the rest of the country or neighbouring countries?

Response: The medicinal uses of the plant were considered to select it for analysis of phytochemicals and screening for biological activity to reveal its potential on wet lab bases, moreover the difference in metabolomics and biological activities may not be huge if the plant is growing in natural habitat.

I believe that a comparative comment on the relevance of the different plants in world or elsewhere's places would be good.

Response: We agree with your suggestion, however we collected the given data for the first time on this specie, so your suggestion will be considered for designing new project for the comparison of same specie growing in different habitats which will be helpful to select the habitat providing most potent medicinal plant

Botanical studies have considerably increased in recent years:

A.M.A. Kawarty, A. M., Behçet, L. & Cakilcioğlu, U. An ethnobotanical survey of medicinal plants in Ballakayati (Erbil, North Iraq). Turk. J. Botany 44, 345–357 (2020).

Satıl, F. & Selvi, S. Ethnobotanical Features of Ziziphora L. (Lamiaceae) Taxa in Turkey. International Journal Nature and Life Sciences. 4, 56-65. (2020).

 Herbs are used for medical purposes. That herbs are used for medical purposes may be added in the introduction part of the article.

Response: Thank you so much for providing valuable and relevant data of medicinal herbs we incorporated it as suggested.

The paper should be edited according to the writing rules of the journal.

Response: MS improved

The references do not conform with the journal style.

Response: The references in the MS are now according to journal style

Reviewer 2 Report

This manuscript summarizes the results of the biological properties of Pleurospermum candollei. Unfortunately, data presentation is imprecise, difficult to follow, and confusing in many parts.

Major comments

  1. The authors used methanol (80%) to prepare the crude extract because phenolic and flavonoid compounds can efficiently be extracted using methanol solvent, as proved by literature. However, the authors also mentioned that this study helps rationalize the use of P. candollei as a nutraceutical substance. Why did the author not extract P. candollei with water or ethanol?
  2. Line 131, the authors mentioned that GC-MS analysis is preferable for non-polar and volatile compounds, including methanolic extract and n-hexane fraction. However, the authors should show the phytochemical compounds of P. candollei chloroform fraction and candollei n-butanol fraction.
  3. In table 4, the authors should show antioxidant activities in terms of half maximal inhibitory concentration (IC50) or half maximal effective concentration (EC50).
  4. The authors should rearrange the table’s heading in tables 4, 5, 7, and 8.
  5. Line 242, the authors should briefly mention the α-amylase and α-glucosidase functions. The authors should link each enzyme.
  6. What is the difference between antioxidant activities in table 4?
  7. The authors also tested thrombolytic activity. How does this test relate to other tests on biological activities topic?
  8. The authors mentioned the relationship between COVID-19 and thrombosis. The authors should state this relationship in the introduction part.
  9. Authors should add the sample size (n) in all table legends.

Author Response

Response to Respected Reviewer 2 comments

Article: “Comprehensive Phytochemical Profiling, Biological activities, and Molecular Docking Studies of Pleurospermum candollei: An Insight into Potential for Natural Products Development".

Thank you for your kind and very keen observation. Also, thank you so much for your valuable insight on improving the manuscript. We greatly appreciate your comments and constructive suggestions. Those valuable comments are constructive to improve our manuscript and provide meaningful guidance for our future research. We hope this revised manuscript will meet your esteemed satisfaction. If you have further suggestion for the improvement of our manuscript, you are always welcome.

  1. The authors used methanol (80%) to prepare the crude extract because phenolic and flavonoid compounds can efficiently be extracted using methanol solvent, as proved by literature. However, the authors also mentioned that this study helps rationalize the use of P. candollei as a nutraceutical substance. Why did the author not extract P. candollei with water or ethanol?

Response: As mentioned we were interested in the bioactive compounds and water/ethanol cannot extract the majority of them. We also mentioned that the “data may help to rational the traditional uses”. However we picked your direction to design the future project to develop nutraceutical formulations, where the extraction will be considered with water/ethanol. Moreover the safety study on the prepared extract may also resolve the concerned issue to design nutraceuticals.

2. Line 131, the authors mentioned that GC-MS analysis is preferable for non-polar and volatile compounds, including methanolic extract and n-hexane fraction. However, the authors should show the phytochemical compounds of P. candollei chloroform fraction and candollei n-butanol fraction.

Response: As the metahol extract contained polar and non-polar compounds and n-hexane extract contain non-polar compounds so both were considered for GCMS analysis, however the chloroform and n-butanol fractions have polar compounds and may not contain fats. So the LCMS will be availed to study the phytochemical profile of these fractions.

3. In table 4, the authors should show antioxidant activities in terms of half maximal inhibitory concentration (IC50) or half maximal effective concentration (EC50).

Response: Kind suggestions are included in table four to improve the MS according to your advice.

4. The authors should rearrange the table’s heading in tables 4, 5, 7, and 8.

Response: MS improved as advised.

5. Line 242, the authors should briefly mention the α-amylase and α-glucosidase functions. The authors should link each enzyme.

Response: MS improved according to the suggestion

6. What is the difference between antioxidant activities in table 4?

Response: Now we categorized the activities to differentiate among them, we also admit the unclear significance in the look of table values, so we added the letters of significance to show a clear difference among the values.

7. The authors also tested thrombolytic activity. How does this test relate to other tests on biological activities topic?

Response: MS improved with reference that the infections may cause clot which leads to the development of other disease. So antibacterial activity may relate to the thrombolytic activity in our study.

8. The authors mentioned the relationship between COVID-19 and thrombosis. The authors should state this relationship in the introduction part.

Response: We added the required information in the introduction part. As we were in sighting into the potential health benefits by screening the biological activities, however more deep study may be designed for the deep analysis of antiviral and thrombolytic effects.

9. Authors should add the sample size (n) in all table legends

Response: MS improved as per your advice.

Reviewer 3 Report

Overall the manuscript is well written and easy to follow. The presentation of the results are clear and straightforward. The English language can be slightly improved to make the manuscript a little more formal.

I have only minor suggestions:

  •  Please  indicate how many replicates were performed for each assay. This  information should  be included in the  Methods section and in the  footnote of each corresponding  table.

  • Page 9 line 165 - prevent the “oxidative” damage

  • For the molecular docking assay  against tyrosine kinase enzyme, why were these ten  compounds specifically selected? Please briefly describe.

  • Page 20 line 545 - please revise the conclusion to “The purpose of current work is to quantify  and  characterize the phytochemical composition and biological activities from the methanolic and different extraction  fractions of P. candollei.” Technically speaking, the purpose of the study was not to determine “the health benefits.”

Author Response

Response to Respected Reviewer 3 comments

Overall the manuscript is well written and easy to follow. The presentation of the results are clear and straightforward. The English language can be slightly improved to make the manuscript a little more formal.

Thank you for your kind and very keen observation for improvement of MS. The authors are extremely thankful for encouraging and appreciating our work and MS.  The authors have improved methodology part where required and also tried to improve English language according to your kind suggestions. We hope this revised manuscript will meet your valued satisfaction. If you have further queries for the improvement of our manuscript, we always welcome your suggestion.

I have only minor suggestions:

  • Please indicate how many replicates were performed for each assay. This information should be included in the Methods section and in the footnote of each corresponding table.

Response: MS improved by including this necessary information where it required.

  • Page 9 line 165 - prevent the “oxidative” damage

Response: Typographical mistake removed in the light of your valuable suggestion.

  • For the molecular docking assay against tyrosine kinase enzyme, why were these ten compounds specifically selected? Please briefly describe.

Response: The selected compounds for molecular docking were sharing the major portion of the extract as estimated from the area of peaks (selected compounds were contributing more than 50% of total peak area) in chromatogram.

  • Page 20 line 545 - please revise the conclusion to “The purpose of current work is to quantify and characterize the phytochemical composition and biological activities from the methanolic and different extraction fractions of P. candollei.” Technically speaking, the purpose of the study was not to determine “the health benefits.

Response: Regarding your technical suggestion the conclusion is revised according to your expert advice.

Round 2

Reviewer 2 Report

-

Author Response

The authors are highly thankful you so much for your valuable insight on improving the manuscript. We hope this revised manuscript will meet the satisfaction of the reviewer.

The authors are  also thankful for encouraging our work. 

Thank you for the conscientious review